# A Comparative Study of Skimmed Milk and Cassava Flour on the Viability of Freeze-Dried Lactic Acid Bacteria as Starter Cultures for Yogurt Fermentation

**DOI:** 10.3390/foods12061207

**Published:** 2023-03-12

**Authors:** Iddrisu Ibrahim, Joseph Atia Ayariga, Junhuan Xu, Robertson K. Boakai, Olufemi S. Ajayi, James Owusu-Kwarteng

**Affiliations:** 1The Microbiology Program, College of Science, Technology, Engineering, and Mathematics (C-STEM), Alabama State University, Montgomery, AL 36104, USA; 2The Industrial Hemp Program, College of Science, Technology, Engineering, and Mathematics (C-STEM), Alabama State University, Montgomery, AL 36104, USA; 3Department of Food Science and Technology, School of Agriculture and Technology, University of Energy and Natural Resources, Sunyani P.O. Box 214, Ghana

**Keywords:** lactic acid bacteria, freeze-drying, yogurt starter culture, fermentation, skimmed milk, cassava flour, cryoprotectants, food, refrigeration, lyophilization

## Abstract

The purpose of this study was to evaluate the survival rates and fermentation performance of three freeze-dried lactic acid bacterial cultures previously isolated from Ghanaian traditional fermented milk. LAB cultures, i.e., *Lactobacillus delbrueckii*, *Lactococcus lactis* and *Leuconostoc mesenteroides*, were frozen in the chamber of a Telstar (Lyoquest) laboratory freeze dryer for 10 h at −55 °C (as single and combined cultures) using skimmed milk and cassava flour as cryoprotectants held in plastic or glass cryovials. For viability during storage, freeze-dried LAB cultures were stored in a refrigerator (4 °C) and at room temperature (25 °C) for 4 weeks. The survival of freeze-dried cultures was determined by growth kinetics at 600 nm (OD_600_). The performance of freeze-dried LAB cultures after 4 weeks of storage was determined by their growth, acidification of milk during yogurt fermentation and consumer sensory evaluation of fermented milk using a nine-point hedonic scale. The survival rates for LAB ranged between 60.11% and 95.4% following freeze-drying. For single cultures, the highest survival was recorded for *Lactobacillus delbrueckii* (L12), whereas for combined cultures, the highest survival was observed for *Lactococcus lactis* (L3) combined with *Lactobacillus delbrueckii* (L12). The consumer acceptability results showed that yogurts produced from a combined starter culture of *Lactococcus lactis* and *Lactobacillus delbrueckii* or from a single culture of *Lactococcus lactis* were the most preferred products with *Lactococcus lactis* and *Lactobacillus delbrueckii* possessing high survival rates and high consumer acceptability in yogurt production. These findings are crucial and can be adopted for large-scale production and commercialization of yogurt.

## 1. Introduction

Fermentation plays a significant role in the traditional processing of food in many parts of the world. In many developing countries, traditional fermentation also serves as the method for improving the shelf life of many staple foods, whilst improving their digestibility, nutritional qualities, organoleptic properties and the degradation of toxins and antinutritive factors [1,2,3,4]. However, in Ghana and many parts of Africa, traditional fermentation processes are natural, that is, without the use of starter cultures (back slopping), which has implications for the safety and quality of the products [5,6,7,8].

In Ghana, naturally fermented milk is commonly produced and consumed by people living in cattle-rearing communities [9,10,11,12]. The production of traditional yogurt-like milk products in Ghana does not rely on the use of commercial starter cultures. Therefore, stocks of previous ferments, fermentation containers and environmental microorganisms contaminating the raw material often initiate fermentation in new batches [13]. During the process, raw or unpasteurized milk is kept in calabashes or plastic containers, covered with a lid, and allowed to spontaneously ferment at ambient temperature (28–35 °C) for about 18–24 h. This natural fermentation results in the formation of curdled milk, yielding a slightly sour yogurt-like product with a pH of less than four with varying consistency [9,10,11,12,13].

Generally, the dependence on such an undefined and diverse microbial consortium during fermentation results in products with inconsistent quality and stability [14,15]. In a framework to develop starter cultures for the controlled fermentation and production of fermented yogurt-like milk with greater consistency in quality and safety, it is required that LAB cultures for commercial and consistent fermentation processes are adequately propagated and made available as concentrates, either in a frozen or freeze-dried form [16,17]. Concentration and preservation of LAB starter cultures for food production rely on technologies, which guarantee the long-term delivery of stable cultures in terms of viability and functional activity [18,19,20]. Thus, the preservation technique of the collected bacterial cultures must ensure that the recovered starter cultures perform in the same manner as the originally isolated species [21].

One method that has commonly been used to prepare dried starter cultures for food applications is freeze-drying [22,23,24]. In this process, the dehydration of LAB imposes environmental stress on the bacterial cells, such as freezing, drying, long-term exposure to low-water activities and rehydration. Microbial survival during this process depends on many factors, including the intrinsic resistance traits of the strains, initial concentration of microorganisms, growth conditions, drying medium and protective agents, storage conditions (temperature, atmosphere, relative humidity) and rehydration conditions [25,26,27,28]. Currently, there are few or no studies that have reported on the survival performance of freeze-dried lactic acid bacteria cultures isolated from traditional Ghanaian fermented milk. Previous investigations on Ghanaian fermented milk products have focused on the isolation and characterization of the predominant microorganisms to develop starter cultures for improved fermentation, food safety and quality [9,10,11,12,13]. The purpose of this study, therefore, was to evaluate the survival rates and fermentation performance of indigenous lactic acid bacterial cultures, isolated from traditional Ghanaian fermented milk, following freeze-drying. Furthermore, the effects of different drying mediums (skimmed milk and cassava floor) and storage conditions (ambient and refrigeration) on the survival and performance of the freeze-dried LAB cultures were also determined.

## 2. Materials and Methods

### 2.1. Study Design

A schematic representation of this study is shown in Figure 1. Previously isolated identified lactic acid bacterial strains or their combinations were treated with cassava flour and skimmed milk as cryoprotectants for freeze-drying. Following freeze-drying, the cultures were stored in plastic or glass containers and stored under different temperatures (4 °C and 25 °C) over four weeks and monitored for their survival rates. Furthermore, the cultures were assessed for their fermentation performance in yoghurt production.

### 2.2. Lactic Acid Bacterial Strains

The strains used in this study include *Lactococcus lactis (L3), L. delbrueckii (L12), Leuconostoc mesenteroides* (L20) and their combinations (L3 + L12, L3 + L20, L12 + L20, L3 + L12 + L20). The strains were isolated from spontaneously fermented milk obtained from Navrongo and Accra in Ghana. The LAB strains were previously identified by the sequencing of the 16S rRNA and in MRS broth with 20% glycerol at −20 °C as stock cultures.

### 2.3. Preparation of Cultures for Freeze Drying

Lactic acid bacterial cells were separately grown in 200 mL MRS broth in Erlenmeyer flasks at 35 °C anaerobically for 24 h. The cells were then harvested by centrifugation (Labofuge200) at 5000 rpm for 10 min. The harvested cells were washed in phosphate buffer solution (PBS) and the initial concentration (Optical Density, OD_600_) of the cells was determined using a spectrophotometer (SM22 PC, SurgienField instrument, Springfield, UK England).

### 2.4. Preparation of Freeze-Drying Career Materials

Skimmed milk powder (20 g) as an excipient was reconstituted in 100 mL of distilled water and autoclaved at 121 °C for 15 min. Cassava flour (20 g) was oven sterilized at 160 °C for 30 min before being mixed with 100 mL distilled water. Reconstituted excipients were allowed to cool to room temperature. The harvested lactic acid bacterial cells were suspended in the reconstituted excipient solutions and aliquoted (500 µL) into 1.8 mL glass and plastic vials.

### 2.5. Freeze-Drying Procedure

LAB samples in glass and plastic vials were frozen in the chamber of a Telstar (Lyoquest) laboratory freeze dryer for 10 h at −55 °C. Vial caps were removed after the freezing was completed and replaced with 100% PTFE thread seal tape (ISO CERTIFIED 19 mm × 0.10 mm × 20 mm) procured from Navrongo market and holes were made on the seal using a sterilized needle procured from Navrongo market.

The LAB samples were loaded in a 500 mL Erlenmeyer flask supported with cotton on the inside and held at the manifolds of the freeze-dryer and subsequently dried using the same Telstar laboratory freeze-dryer under a vacuum pressure of 100 mtor (1.33 mbar) for 20 h. LAB samples were disconnected from the freeze-dryer and the sealed tapes were immediately replaced with the caps of the cryovials to avoid contamination. Cell viability was measured immediately after freeze-drying and this represents the initial (Day 0) of cell viability.

### 2.6. Storage Conditions for Freeze-Dried Cultures

The LAB samples were divided into two equal portions, with each portion containing equal samples held in plastic and glass vials. Half of the LAB samples were stored in refrigerator (refrigerated condition) and the other half was stored at room temperature (ambient condition). A weekly viability study was conducted on the stored LAB samples for four weeks.

### 2.7. Viability Assays

#### 2.7.1. Preparation of Resuscitation Media

Freeze-dried samples (10 mg) were each reconstituted in 900 µL of peptone water and vortexed to obtain a uniform mix, and 100 µL each of the reconstituted samples was then added to 5 ml of prepared MRS broth and incubated at 35 °C for 24 h.

#### 2.7.2. Measurement of Optical Density (OD)/Viable Cell Counting

Initial cell concentration (optical density) was determined using a spectrophotometer (SM22 PC, SurgienField instrument, Springfield, UK England) for cell viability and survivability at a wavelength of 600 nm (OD_600_). A plastic reusable cuvette was used to hold the samples to be measured in the spectrophotometer machine. The cuvette was wiped with clean cotton soaked in ethanol before the next measurement was taken to avoid cross-contamination. OD_600_ values were measured after the first week of storage, and the experiment was repeated three more times (once every week).

#### 2.7.3. Determination of Survival Rates

The percentage survival of the strains after the freeze-drying process was expressed as follows:Survival % = N_i_/N_f_ × 100
where N_f_ is the CFU/g at the end of freeze-drying and N_i_ is the CFU/g before freeze drying (at the end of centrifugation) [29].

### 2.8. Performance Evaluation of Freeze-Dried Culture in Milk Fermentation

#### Inoculation of Cow Milk Samples and Fermentation

Freshly collected cow milk samples were standardized to have 4% fat and distributed into 250 mL Erlenmeyer flasks at 100 mL per flask. The milk samples were then pasteurized at 75 °C for 15 min and cooled to 35 °C [30]. Milk samples were inoculated with 0.01 % (*w*/*v*) of freeze-dried cultures (having a viable count of approximately 10^8^ CFU/mL) according to the single and combined starter cultures shown in Table 1. Inoculated milk samples were incubated at 35 °C for 16 h [31]. For spontaneous fermentation (control), fresh milk was allowed to ferment in a clean plastic container at ambient temperature for 16 h without initial pasteurization of the milk.

### 2.9. Bacterial Growth and Acidification of Milk

For the determination of bacterial growth during starter culture fermentation, 10 mL of fermenting samples were serially diluted (10^−1^ to 10^−9^) using a sterile phosphate buffer solution. Appropriate decimal dilutions were spread on an MRS agar. After solidification, inoculated plates were incubated at 30 °C for 48 h. The colonies were counted and expressed as log_10_ CFU/mL.

To determine the performance of freeze-dried bacterial cultures by acidification properties, the pH of fermenting milk samples was determined using a digital pH meter (Crison basic 20, Barcelona), calibrated with standard buffer solutions at 30 (±2) °C. All measurements were carried out in triplicate and presented as means ± standard deviations. Furthermore, titratable acidity (TA) was determined by titrating fermenting milk samples against 0.1 N sodium hydroxide (NaOH) solution and the results were expressed as % lactic acid produced.

### 2.10. Consumer Sensory Evaluation of Fermented Milk

The fermented milk products (yogurt) prepared by fermentation with different freeze-dried starter cultures were served to 50 volunteers of untrained panelists (drawn from the University for Development Studies and Navrongo Community) who are familiar with the traditional yogurt [32]. In separate sensory evaluation booths, the panels independently evaluated the various products for their sensory qualities including color, odor, taste, texture and overall acceptability using a nine-point hedonic scale with one, five and nine representing ‘dislike extremely’, ‘neither like nor dislike’ and ‘like extremely’, respectively. All six fermented milk products were presented to the panelists randomly placed side-by-side, with each panelist receiving two rounds of each product and water for rinsing their mouths. The spontaneously fermented milk (without added known starter culture) was served as the control sample. Before the assessment, a detailed explanation of the process of evaluation was given to the panelists. After judging appearance, the panelists were then allowed to taste the samples and evaluate other sensory properties using the nine-point hedonic scale. The assessors were made to wash their mouths with water after evaluating each product.

### 2.11. Statistical Analysis

All experiments were repeated at least three times and raw data generated were entered into excel spreadsheet for further processing and management. Descriptive statistics including bar charts and line (time-series) graphs were used to analyze survival rates and performance of the freeze-dried LAB cultures. One-way analysis of variance (ANOVA) was used to compare the means. The means were separated by Tukey’s family error rate multiple comparison test using the MINITAB statistical software package (MINITAB Inc. Release 14 for windows, 2004), and the differences in means were considered statistically significant at *p* < 0.05.

## 3. Results

### 3.1. Viability of Single Strains Lactic Acid Bacteria after Freeze Drying

Single strains of lactic acid bacteria *Lactococcus lactis* (L3), *Lactobacillus delbrueckii* (L12) and *Leuconostoc mesenteroides* (L20) were freeze-dried in two different kinds of excipients (cassava—CSA and skimmed milk—MLK) and in two different kinds of storage material (plastic—P and glass—G) to ascertain which treatment/condition best suits or supports the functionality and viability of the lactic acid bacteria after a period of four weeks following freeze-drying. The results showed that both *Lactococcus lactis* (L3) (Figure 2) and *L. delbrueckii* (L12) (Figure 3) survived very well in cassava using plastic as the storage material (CSA_P) representing 68.87% and 70.91%, respectively, and *Leuconostoc mesenteroides* (L20) (Figure 3) survived well in skimmed milk under glass as the storage (MLKG) material representing 68.36% immediately after freeze drying (control).

*L. delbrueckii (L12)* survived highly (Figure 3) (70.91%) when treated with cassava with plastic as the storage container as compared to the other cultures immediately after freeze drying (Figure 3) (Control).

### 3.2. Viability of Single Strains Lactic Acid Bacteria following Freeze-Drying after Weeks of Storage

*Lactococcus lactis* (L3) showed superior survival rates across all the weeks (4) of storage in all the different treatments as it recorded a survival rate of between 55.6% and 95.4% under both storage temperatures. At the end of the fourth week, L3 survived better in skimmed milk using glass as the storage container representing a 76.6% survival rate under a refrigerated storage temperature of 4 °C (Figure 2A) after 48 h storage time and 76.6% after 24 h storage time. Cassava in plastic least supported the survival of L3 at the end of the fourth week, representing 71.8% (Figure 2A) in refrigerated conditions. Though ambient conditions (25 °C) least supported the survival rates of L3 at the end of the fourth week, L3 still survived above 50 across all the treatments (Figure 2B).

The survival rate of *Lactobacillus delbrueckii* (L12) ranges between 57.0% and 89.3% across all the weeks of storage in all the different treatments under refrigerated temperature, and 58.7% to 87% at ambient temperatures (Figure 3A,B). L12 survived better in the first week (89.3%) when treated with cassava under the plastic storage container and survived better at the end of the fourth week when treated in skimmed milk using the glass storage container, representing a 78.8% survival rate under a refrigerated storage temperature of 4 °C (Figure 3A).

The survival rate of *Leuconostoc mesenteroides* (L20) ranges between 56.6% and 92% across all the weeks of storage in all the different treatments under ambient temperatures (4 °C) and 56.6% to 82.6% under an ambient storage temperature (25 °C) (Figure 4A,B). However, L20 survived better at the end of the fourth week when treated with cassava using the plastic storage container, representing a 92% survival rate under the refrigerated storage temperature of 4 °C and a 63.4% survival rate when treated under the glass storage container at an ambient temperature of 4 °C (Figure 4A,B)**.**

### 3.3. Viability of Combined Strains Lactic Acid Bacteria after Freeze-Drying After

Combined strains of lactic acid bacteria (*Lactococcus lactis* (L3) + *Lactobacillus delbrueckii* (L12), *Lactococcus lactis* (L3) + *Leuconostoc mesenteroides* (L20), *Lactobacillus delbruesckii* subsp bulgaricus (L12) + *Leuconostoc mesenteroides* (L20) and *Lactococcus lactis* (L3) + *Lactobacillus delbrueckii* (L12) + *Leuconostoc mesenteroides* (L20)) were also freeze-dried in two different kinds of excipients (cassava—CSA and skimmed milk—MLK) and in two different kinds of storage material (plastic—P and glass—G) to ascertain which treatment best supports the functionality of the lactic acid bacteria following freeze drying. Results showed that all the different combined strains survived very well in cassava using plastic as the storage material (CSA_P) with *Lactococcus lactis* (L3) + *Lactobacillus delbruesckii* (L12) having the highest survival rates ranging from 60.0% to 70.7%, *Lactococcus lactis* (L3) + *Leuconostoc mesenteroides* (L20) representing 67.3%, *Lactobacillus delbrueckii* (L12) + *Leuconostoc mesenteroides* (L20) representing 68.21% and *Lactococcus lactis* (L3) + *Lactobacillus delbruesckii* subsp bulgaricus (L12) + *Leuconostoc mesenteroides* (L20) representing 68.19%. However, *Lactococcus lactis* (L3) + *Lactobacillus delbrueckii* (L12) survived better (69.84%) when treated with cassava with plastic as the storage container (A) immediately after freeze drying (control) (Figure 5, Figure 6 and Figure 7).

### 3.4. Viability of Combined Strains Lactic Acid Bacteria following Freeze Drying after Weeks of Storage

The survival rate of the combined cultures of *Lactococcus lactic* (L3) *and Lactobacillus delbrueckii* (L12) ranges between 61.9% and 94.4% across all the weeks of storage under a refrigerated temperature (4 °C) and 55.9% to 84.1% under an ambient temperature (25 °C) in all the different treatments under both storage temperatures (Figure 5A,B). However, L3 + L12 survived superiorly in the first week when treated with cassava under the plastic storage container at a refrigerated temperature of 4 °C (94.4%). L3 + L12 survived better at the end of the fourth week when treated in cassava using the plastic storage container representing an 84.1% survival rate under an ambient storage temperature of 25 °C and a 65.3% survival rate when treated with skimmed milk under the glass storage container at a refrigerated temperature of 4 °C (Figure 5A,B).

The survival rate of the combined cultures *of Lactococcus lactis* (L3) and *Leuconostoc mesenteroides* (L20) ranges between 54.2% and 80.5% under a refrigerated temperature (4 °C) and 61.6% to 80.8% under an ambient temperature (25 °C) across all the weeks of storage in all the different treatments under both storage temperatures (Figure 6A,B). L3 + L20, however, survived better at the end of the fourth week when treated in cassava using the glass storage container representing an 80.5% survival rate under a refrigerated storage temperature of 4 °C and a 68.9% survival rate when treated with skimmed milk under the glass storage container at an ambient temperature of 4 °C (Figure 6A,B).

The survival rate of the combined cultures of *Lactobacillus delbrueckii* (L12) and *Leuconostoc mesenteroides* (L20) ranges between 61.8% and 86.4% under ambient temperatures of 25 °C and 60.0% to 78.7% under refrigeration temperatures of 4 °C across all the weeks of storage in all the different treatments under both storage temperatures (Figure 7A,B). L12 + L20, however, survived better at the end of the fourth week when treated in cassava using the glass storage container representing a 78.5% survival rate under a refrigerated storage temperature of 4 °C and a 78.7% survival rate when treated in cassava in glass at a refrigerated temperature of 4 °C (Figure 7A,B).

The survival rate of the combined cultures of *Lactococcus lactis* (L3), *Lactobacillus delbrueckii* (L12) and *Leuconostoc mesenteroides* (L20) ranges between 56.4% and 83.3% under a refrigerated temperature of 4 °C and 60.0% to 84.6% under an ambient temperature of 25 °C across all the weeks of storage in all the different treatments under both storage temperatures (Figure 8A,B). L3 + L12 + L20, however, survived better at the end of the fourth week when treated with cassava in glass representing 83.3% under a refrigerated temperature of 4 °C and a 70.6% survival rate when treated with cassava in glass under an ambient temperature of 4 °C (Figure 8A,B).

### 3.5. Performance of Freeze-Dried LAB Starter Cultures during Milk Fermentation

For the performance of freeze-dried cultures, the total lactic acid bacterial count and pH were determined during milk fermentation. Generally, the LAB counts increased with fermentation time while the pH decreased (Figure 9). *Lactococcus lactis* recoded the lowest pH rate as well as the highest CFU/mL, while yogurt fermented from the spontaneous fermentation of milk recorded the highest pH value as well as the lowest CFU/mL at the end of the fermentation period.

### 3.6. Consumer Sensory Evaluation of Yogurts Fermented with Freeze-Dried Starter Culture

The effect of freeze-dried LAB cultures on the consumer sensory attribute was evaluated using a nine-point hedonic scale with 1, 5 and 9 and is presented in Table 1. The type of starter culture did not have a significant effect on the color of yogurt. However, a significant difference (*p* < 0.05) was observed among the different starter cultures on yogurt odor, taste, texture and overall acceptability. The combined starter culture of *Lactococcus lactis* (L3), *Lactobacillus delbrueckii* (L12) and a single starter of *Leuconostoc mesenteroides* (L20) were highly scored for odor, taste and texture. For overall acceptability, consumers scored yogurts produced with a combined starter culture of *Lactococcus lactis* (L3) and *Lactobacillus delbrueckii* (L12) or single culture of only *Lactococcus lactis* as the most preferred products. Except color where no significant difference was observed, yogurt produced by spontaneous fermentation was least preferred in all other sensory attributes (Table 2). In addition, there was no significant difference between all the strains (single and combined) including yogurt produced from spontaneous fermentation regarding appearance (color). For product odor, taste as well as texture, the combined strains of L3 + L12 are significantly higher as compared to the other strains, except for L3 which showed no significant difference compared to L3 + L12 regarding taste (*p* < 0.05). Nonetheless, yogurt produced with a single starter culture of *Lactococcus lactis* (L3) or combined starter cultures of *Lactococcus lactis* and *Lactobacillus delbrueckii* (L3 + L12) showed significantly higher overall acceptability.

## 4. Discussion

Lactic acid bacteria are predominantly used in the fermentation of milk into fermented milk products, often referred to as yogurt starter cultures or simply starter cultures [33,34,35,36]. The purpose of this study was to investigate the survival rates of lactic acid bacteria following freeze-drying. Cultures to be freeze-dried were treated with skimmed milk powder and cassava flour, which serve as cryoprotectants/excipients, to compare which cryoprotectants best support the survivability of the cultures. Additionally, we investigated the impact of refrigeration and ambient temperatures exert on the cultures following one-month-long storage. Additionally, *Lactococcus lactis* (L3), Lactobacillus delbrueckii (L12)*,* and *Leuconostoc mesenteroides* (L20) and their combinations, *Lactococcus lactis* (L3) + Lactobacillus delbrueckii subsp bulgaricus (L12), Lactococcus lactis (L3) + Leuconostoc mesenteroides (L20), Lactobacillus delbrueckii subsp bulgaricus (L12) + Leuconostoc mesenteroides (L20) and *Lactococcus lactis* (L3) + Lactobacillus delbrueckii subsp bulgaricus (L12) + Leuconostoc mesenteroides (L20) were the lactic acid bacteria used for this study as they are the most common kind of starter cultures used in yogurt production. [37,38,39,40].

It was observed during the studies that the strains performed better under both refrigerated and ambient storage temperatures in the fourth week. Wirjantoro and Phianmongkhol investigated the survival of three different species of lactic acid bacteria and *Bifidobacterium bifidum* in yogurt powder during 4 weeks of storage at room and refrigerator temperatures. They employed spray-drying technology and they packaged yogurt in PET/PP/Al or nylon/PE packaging. 

Their results revealed a significant reduction in lactic acid bacteria and *B. bifidum* for up to 4.65 log cfu/mL after the drying process. They observed, however, that storing the dried yogurt in refrigerated temperatures generally improved the survival of the desired microorganisms. Their studies further revealed that PET/PP/Al better supported the viability of the organisms compared to nylon/PE except for *L. bulgaricus* [41].

This observation is of immense interest as we are primarily trying to preserve microbial cultures for a longer period (4 weeks in this study) while minimizing cost. Therefore, the ability of the strains to perform well at the ambient condition during the fourth week suggest that within the limits of this study, the strains/cultures may not need refrigeration over 4 weeks’ storage to be used as viable starter cultures. This will reduce to a large extent the cost of maintaining these cultures. In addition, the transportation of cultures will be less hectic as there will be no need to carry cultures under refrigeration. 

Microbial preservation by freeze-drying is known to preserve the viability of cultures over long durations [42]. In this work, freeze drying proved to be effective in achieving high survival rates as all three strains and their combinations achieved >50% survival rates. Our results also agree with those reported by [29], who found very high survival rates after freeze drying the strains of *L. Plantarum* VE36, G2/25 and *L. pentosus* LB61 with survival rates of 97.3%, 79.9% and 76.7% respectively, which they employed in gari fermentation. The high survival rate in this study may be attributed to the quality of the sterilization techniques adopted, the efficiency of the freeze dryer itself and/or the choice of the excipients used (cassava flour and skimmed milk powder) because good excipients/cryoprotectants can greatly impact survival rates of bacteria [43,44]. The impact of cryoprotectants such as glucose, lactose, trehalose and skimmed milk or their combination was assessed by Jofré et al., [45] on the survival of probiotic *Lactobacillus rhamnosus* CTC1679, *Lactobacillus casei/paracasei* CTC1677 and *L. casei/paracasei* CTC1678 during freeze drying and after 39 weeks of storage at 4 °C and 22 °C. They recorded high survival rates (≥94%) immediately after freeze-drying with slight differences observed among strains and cryoprotectants. They also reported that cryoprotectants, temperature and strains impacted the survival of the bacteria during storage and that the stability of the cultures survived highly (reductions ≤ 0.9 log units after 39 weeks of storage) under refrigeration temperatures (4 °C) using skimmed milk alone or combined with trehalose or lactose as cryoprotectants. The lowest survival was observed during non-refrigerated storage with glucose and glucose plus milk as cryoprotectants; no viable cells were left at the end of the storage period. They concluded that freeze drying using appropriate cryoprotectants such as skimmed milk allows for the long shelf-life of highly concentrated dried bacterial cultures.

Not all microorganisms can be successfully freeze-dried, especially mutants with deficient membranes. Several reports have shown lactic acid bacteria have been successfully freeze-dried [29,44,45,46] with different freeze-drying approaches. Miyamoto-Shinohara et al., [47] assessed the survival rates of 10 species of microorganisms after freeze drying and preserving in a vacuum at 5 °C. With varied survival rates recorded as indicated in our study, they stated that the survival rates of *Brevibacterium flavum, B. lactofermentum, Corynebacterium acetoacidophilum, C. gultamicum* and *Streptococcus mutans* were around 80%. The survival rate of *Brevibacterium* and *Corynebacterium* did not decrease greatly during the storage period, whereas the rate of *S. mutans* decreased to about 20% after 10 years. Survival rates after the drying of Gram-negative bacteria, i.e., *Escherichia coli, Pseudomonas putida, Serratia marcescens,* and *Alcaligenes faecalis* were around 50%. The survival rate decreased for the first 5 years and then stabilized at around 10% thereafter.

This may be due to the removal of the most sensitive parts of the cell population during the freeze-drying process and the low surface area of the selected strains. The high survival rates may also be attributed to the rehydration method adopted in this study, as there is an increase in survival rate when the rehydration process is slowed [48]. It may also be due to the high initial cell concentration of the microorganisms, the growth conditions and the growth and resuscitation media; all these conditions affect the viability of bacteria cultures.

Jouki et al. investigated the effects of sorbitol along with sodium alginate-skimmed milk and glycerol as cryoprotectants on the viability of *Lactobacillus plantarum* microcapsules in freeze-dried yogurt. They recorded survival rates of 67.1%, 89.4% and 91.2% for *L. plantarum*, cryoprotected *L. plantarum* microcapsules, and cryoprotected *L. plantarum* microcapsules, respectively [49].

Our studies showed that all three strains and their combinations were not affected greatly by the storage containers as well as the storage temperature as all strains survived above 50%. Yao et al. [29] equally recorded above 50% survival rates of 12 LAB strains out of 16 LAB strains, which they subjected to freeze-drying. The excipients (skimmed milk and cassava flour) were both able to preserve the cells from loss of viability. Our study thus suggests that freeze-dried cultures maybe are stored in glass containers as a very long storage time can cause atmospheric water to diffuse into plastic tubes and damage freeze-dried samples, as suggested by [50]. Furthermore, our studies showed that samples stored in plastic containers showed a very good survival rate over the four-week storage duration. Both excipients (skimmed milk and cassava flour) used in this study supported the viability of the strains. 

However, strains treated with cassava flour showed a superior survival rate over strains treated with skimmed milk, as indicated by the results above. This may be due to cassava being a polysaccharide (exopolysaccharide) with some sort of biological essence. Exopolysaccharides have two types of secreted polysaccharides, with the first type (capsular polysaccharide) attached to the cell wall as a capsule and the second type (slime exopolysaccharide) produced as a loose unattached material [51,52,53]. Exopolysaccharides prevent microbial cells from desiccation, phagocytosis, phage attachment, antibiotics, toxic compounds and osmotic stress [31,54]. Skimmed milk is usually selected when it comes to industrial, scale-up or commercial production of freeze-dried lactic acid bacteria [55,56]. This may be due to several factors such as the prevention of cellular injury by stabilizing the cell membrane, the creation of a porous structure in the freeze-dried product that makes rehydration easier, and finally, it contains proteins that provide a protective coating for the cells [49,50]. This is evident in our study as both excipients led to high survival rates of the LAB strains. 

In addition to temperature, relative humidity and exposure to light all impact the survival of freeze-dried samples [57]. Our studies have shown that the bacteria strains used can survive under both ambient (25 °C) and refrigerated (4 °C) conditions. However, some studies have suggested that freeze-dried samples may be stored in environments with lower temperatures as it impacts survival rates for the long storage of samples [58,59]. Therefore, freeze-dried samples are suggested to be stored in a relatively balanced environment, and samples are recommended to not be stored in temperatures above 30 °C. Samples are also suggested to be stored under a vacuum and exposed to darkness [60].

In the development of starter cultures for the production of fermented milk products, quick acidification is a topmost priority [61]. The acidification rates varied among the freeze-dried LAB strains tested. Akabanda et al. [31] reported that *Lactobacillus helveticus*, *L. fermentum*, *L. plantarum* and *L. mesenteroides* isolated from nunu were the fastest acid producers in comparison to the other strains used in their study. From the results obtained, all the selected strains tested showed a fast rate of acidification during the period of fermentation. A decrease in pH is essential in yogurt production as it accelerates coagulation and mitigation of pathogenic microflora that might invade the products [62,63]. The selected freeze-dried strains are therefore good candidates as starter cultures for the dairy fermentation process. LAB starter cultures that possess the ability to degrade lactose rapidly and completely to lactic acid with minimal nutritional levels are generally desirable [64,65]. Accelerated acidification of the raw materials means the prevention of the growth of undesirable microorganisms on fermented products. This has a positive impact on the aroma, texture and flavor of the end product. A rapid decrease in pH to <4 indicates the fastest growth and inhibition of starter cultures against pathogenic microbes, especially *Salmonella* spp [66].

Higher colony-forming units indicate high viability and vice versa. Even though the LAB cultures survived fairly well at the end of freeze drying and following storage over four weeks, the viability still needed to be confirmed by their ability to ferment milk at the end of the fourth week after freeze drying. Hence, we subjected the LAB to the fermentation of milk (to produce yogurt) for 12 h measuring cell growth and viability every 3 h. In antiquity, consumer sensory evaluation was employed and served as the basis for which foods were accepted or rejected. Consumer sensory evaluation is a technique that examines the qualities of food products through the senses such as smell, hearing, taste, touch and sight of the panelists. Some food quality parameters mostly considered include texture, flavor, smell, taste, appearance, etc. Consumer sensory evaluation acts as a complement to the microbiological and technological methods employed in assessing the safety of foods, especially as a final step of food product evaluation [67,68,69]. We employed this technique in our study to ascertain the acceptable levels of yogurt produced from the freeze-dried cultures compared to the traditional (spontaneous) fermentation. Consumer sensory analysis in this study showed varying degrees of acceptability for yogurt fermented with the different starter cultures. Generally, yogurt fermented with freeze-dried lactic acid bacteria cultures, either single or combined strains from this study, showed improved acceptability as compared to the spontaneously fermented yogurt. The high acceptability of yogurt fermented with *Lactococcus lactis* (L3) and the combined cultures of *Lactococcus lactis* and lactobacillus delbrueckii (L3 + L12) could be due to the cultures being able to reduce the pH of the milk from 6.45–3.18 and 6.44–3.20, respectively. Reduction in pH is critical as it affects the organoleptic properties of the yogurt [70]. This may be because these cultures were able to produce the lowest pH values (high acidity) during the yogurt fermentation at the end of 12 h [71]. Park et al. recorded an unpleasant acid taste with a yogurt acidity of more than 1.8% and a titratable acidity of about 1.15% considered as the average [72]. Sung et al. evaluated the effect of added freeze-dried mulberry fruit juice on the antioxidant activity and fermented properties of yogurt during refrigerated storage. They observed a decrease in the pH of yogurt and an increase in acidity during fermentation. They also observed that the initial lactic acid bacteria count of yogurt was reduced from 6.49–6.94 Log CFU/g and increased above 9 Log CFU/g in control and 1% in freeze-dried mulberry fruit juice yogurt for 24 h. In sensory evaluation, yogurt with reduced pH was ranked higher when compared with other yogurts [70]. 

## 5. Conclusions

In this study, the potential of the three pre-selected LAB cultures (*Lactococcus lactis, Lactobacillus delbrueckii* and *Leuconostoc mesenteroides*) for use as suitable freeze-dried starter cultures for milk fermentation during yogurt production was assessed. In general, survival rates for lactic acid bacterial cultures ranged between 60.11% and 70.91% following freeze-drying. For single cultures, the highest survival was recorded for *Lactobacillus delbrueckii* (L12), whereas for combined the highest survival was observed for *Lactococcus lactis* (L3) combined with *Lactobacillus delbrueckii* (L12). The different excipients (cassava and milk) have different varying effects on the different bacterial cultures. All freeze-dried lactic acid bacteria starter cultures whether single or in combinations that were selected after the freeze drying grew rapidly during yogurt making, reducing the pH of milk to below four units within 9 h of fermentation. On the other hand, spontaneous fermentation (without starter cultures) was characterized by slower acidification. For overall consumer acceptability, yogurts produced with a combined starter culture of *Lactoccus lactis* and *Lactobacillus delbrueckii* or single culture of *Lactococcus lactis* were the most preferred products.

Overall, *Lactococcus lactis* and *Lactobacillus delbrueckii* can be used as freeze-dried lactic acid bacterial starter culture with high survival rates and high consumer acceptability in the production of yogurt. This can be adopted for large-scale production and commercialization of yogurt production. Notwithstanding, further studies on the effects of freeze drying and long-term storage on the survival and performance of selected LAB cultures are recommended.

## Figures and Tables

**Figure 1 foods-12-01207-f001:**
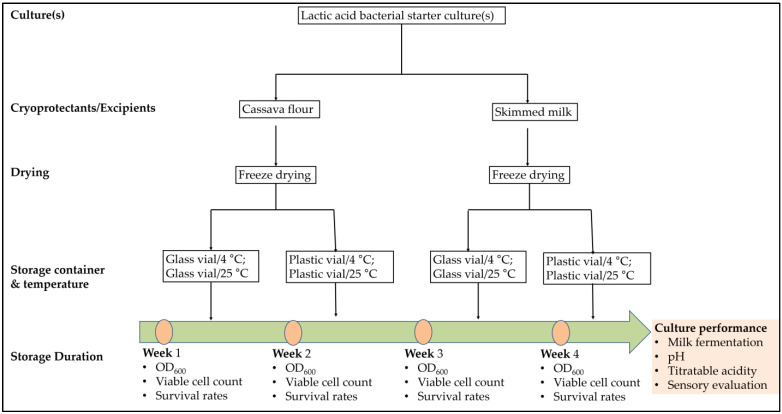
Study design; each selected lactic acid bacteria: *Lactococcus lactis* (L3), *L. delbrueckii* (L12), *Leuconostoc mesenteroides* (L20) and their combinations (L3 + L12, L3 + L20, L12 + L20, L3 + L12 + L20) go through these treatments.

**Figure 2 foods-12-01207-f002:**
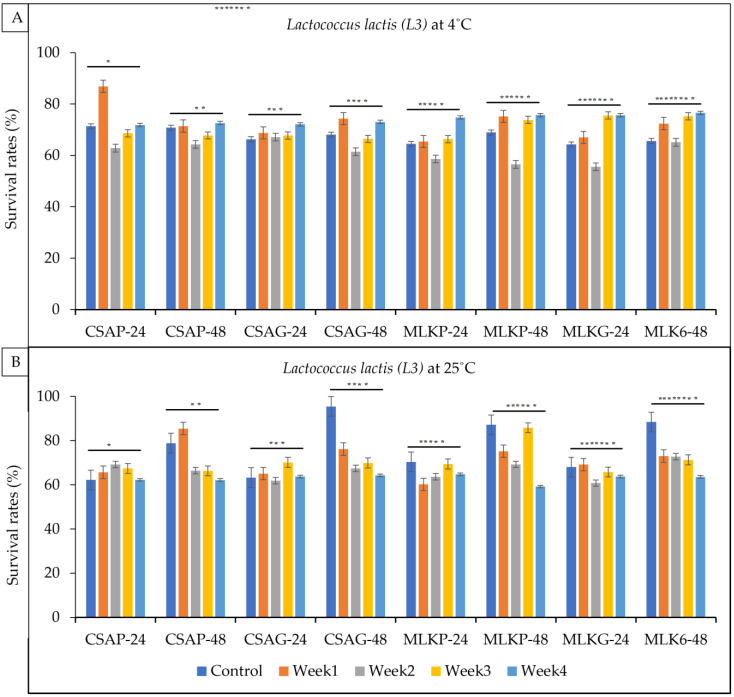
Viability of *Lactococcus lactis* (L3) following freeze-drying. (**A**) Survival rates at ambient temperature (25 °C), and (**B**) survival rates at refrigerated temperature (4 °C) for 24 h and 48 h. CSAP-24-Cassava in plastic at 24 h, CSAP-48-Cassava in plastic at 48 h, CSAG-24-Cassava in glass at 24 h, CSAG-48-Cassava in glass at 48 h, MLKp-24-Milk in plastic at 24 h, MLKP-48-Milk in plastic at 48 h, MLKG-24-Milk in glass at 24 h, MLKG-48-Milk in glass at 48 h. (Values represent means of three replicate experiments, ±: standard deviation). For (**A**) * *p*-value = 0.47081, ** *p*-value = 0.19330, *** *p*-value = 0.02660, **** *p*-value = 0.42210, ***** *p*-value = 0.22985, ****** *p*-value = 0.45262, ******* *p*-value = 0.12782, ******** *p*-value = 0.00020. For (**B**) * *p*-value = 0.00015, ** *p*-value = 0.00209, *** *p*-value = 0.051444, **** *p*-value = 0.08801, ***** *p*-value = 0.00031, ****** *p*-value = 0.00046, ******* *p*-value = 0.00290, ******** *p*-value = 0.00001.

**Figure 3 foods-12-01207-f003:**
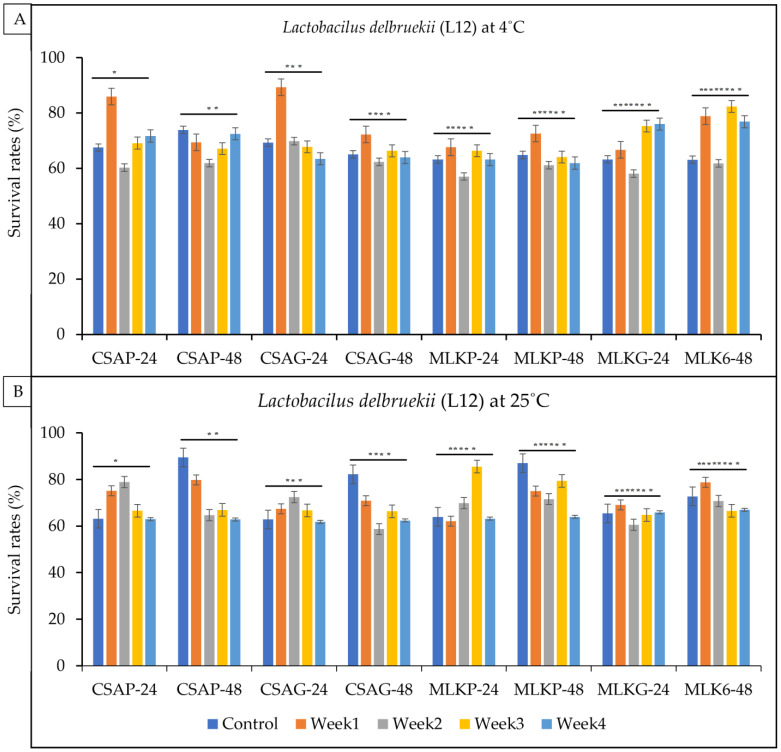
Viability of *Lactobacillus delbrueckii* (L12) following freeze-drying. (**A**) Survival rates at ambient temperature (25 °C), and (**B**) survival rates at refrigerated temperature (4 °C) for 24 h and 48 h. CSAP-24-Cassava in plastic at 24 h, CSAP-48-Cassava in plastic at 48 h, CSAG-24-Cassava in glass at 24 h, CSAG-48-Cassava in glass at 48 h, MLKp-24-Milk in plastic at 24 h, MLKP-48-Milk in plastic at 48 h, MLKG-24-Milk in glass at 24 h, MLKG-48-Milk in glass at 48 h. (Values represent means of three replicate experiments, ±: standard deviation). For (**A**) * *p*-value = 0.12298, ** *p*-value = 0.00031, *** *p*-value = 0.26069, **** *p*-value = 0.22889, ***** *p*-value = 0.46894, ****** *p*-value = 0.43324, ******* *p*-value = 0.02103, ******** *p*-value = 0.00023. For (**B**) * *p*-value = 0.00076, ** *p*-value = 0.000001, *** *p*-value = 0.00436, **** *p*-value = 0.0000001, ***** *p*-value = 0.02743, ****** *p*-value = 0.00001, ******* *p*-value = 0.31043, ******** *p*-value = 0.07355.

**Figure 4 foods-12-01207-f004:**
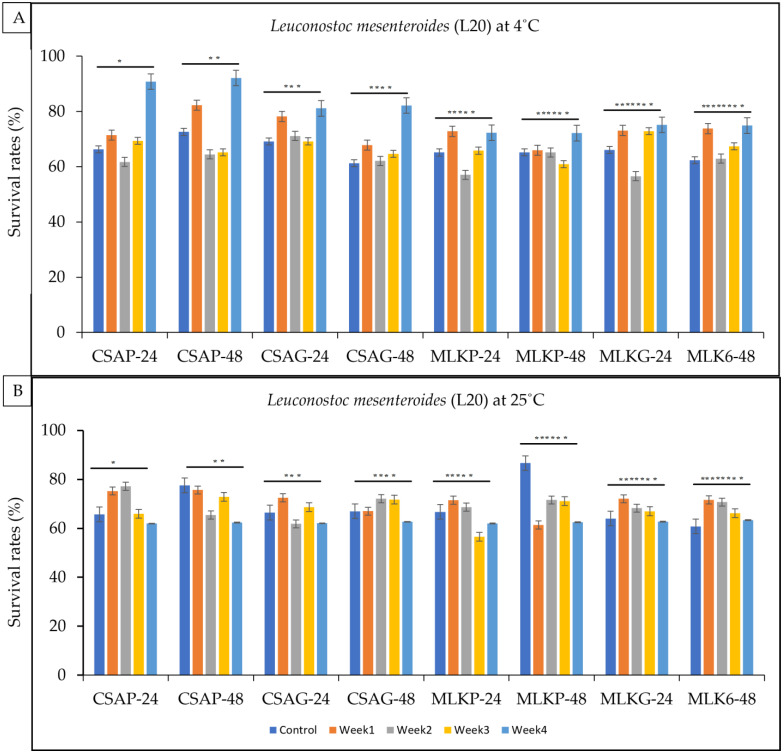
Viability of *Leuconostoc mesenteroides* (L20) following freeze-drying. (**A**) Survival rates at ambient temperature (25 °C), and (**B**) survival rates at refrigerated temperature (4 °C) for 24 h and 48 h. CSAP-24-Cassava in plastic at 24 h, CSAP-48-Cassava in plastic at 48 h, CSAG-24-Cassava in glass at 24 h, CSAG-48-Cassava in glass at 48 h, MLKp-24-Milk in plastic at 24 h, MLKP-48-Milk in plastic at 48 h, MLKG-24-Milk in glass at 24 h, MLKG-48-Milk in glass at 48 h. (Values represent means of three replicate experiments, ±: standard deviation). For (**A**) * *p*-value = 0.03162, ** *p*-value = 0.30846, *** *p*-value = 0.00104, **** *p*-value = 0.00153, ***** *p*-value = 0.35890, ****** *p*-value = 0.30947, ******* *p*-value = 0.16470, ******** *p*-value = 0.00014. For (**B**) * *p*-value = 0.00081, ** *p*-value = 0.00025, *** *p*-value = 0.36361, **** *p*-value = 0.16799, ***** *p*-value = 0.09687, ****** *p*-value = 0.0000001, ******* *p*-value = 0.00267, ******** *p*-value = 0.000004.

**Figure 5 foods-12-01207-f005:**
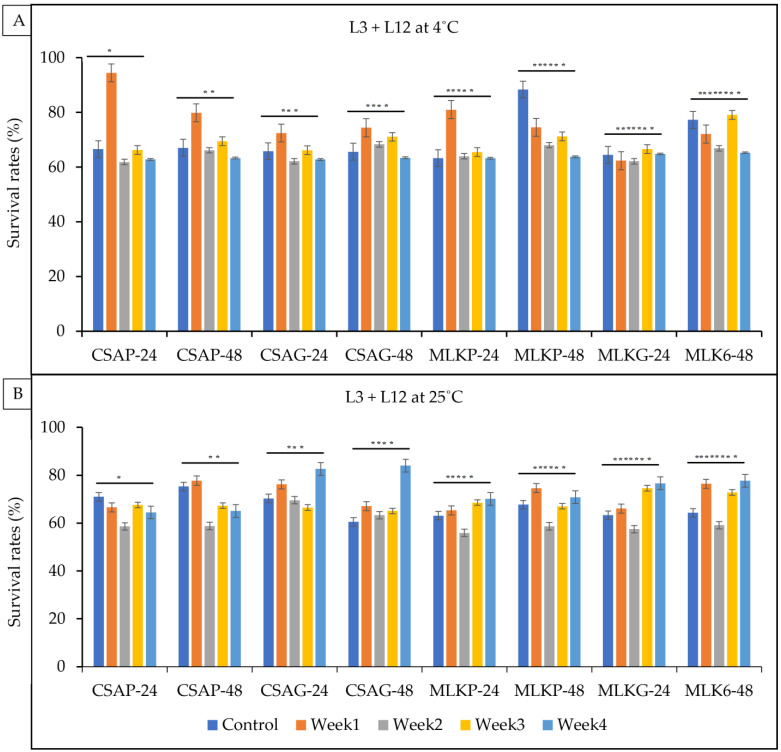
Viability of *Lactococcus lactis* (L3) + *Lactobacillus delbrueckii* (L12) following freeze-drying. (**A**) Survival rates at ambient temperature (25 °C), and (**B**) survival rates at refrigerated temperature (4 °C) for 24 h and 48 h. CSAP-24-Cassava in plastic at 24 h, CSAP-48-Cassava in plastic at 48 h, CSAG-24-Cassava in glass at 24 h, CSAG-48-Cassava in glass at 48 h, MLKp-24-Milk in plastic at 24 h, MLKP-48-Milk in plastic at 48 h, MLKG-24-Milk in glass at 24 h, MLKG-48-Milk in glass at 48 h. (Values represent means of three replicate experiments, ±: standard deviation). For (**A**) * *p*-value = 0.21514, ** *p*-value = 0.12357, *** *p*-value = 0.44616, **** *p*-value = 0.00959, ***** *p*-value = 0.03607, ****** *p*-value = 0.000000, ******* *p*-value = 0.314714, ******** *p*-value = 0.00102. For (**B**) * *p*-value = 0.000060, ** *p*-value = 0.00127, *** *p*-value = 0.06336, **** *p*-value = 0.00053, ***** *p*-value = 0.22466, ****** *p*-value = 0.40781, ******* *p*-value = 0.05309, ******** *p*-value = 0.00786.

**Figure 6 foods-12-01207-f006:**
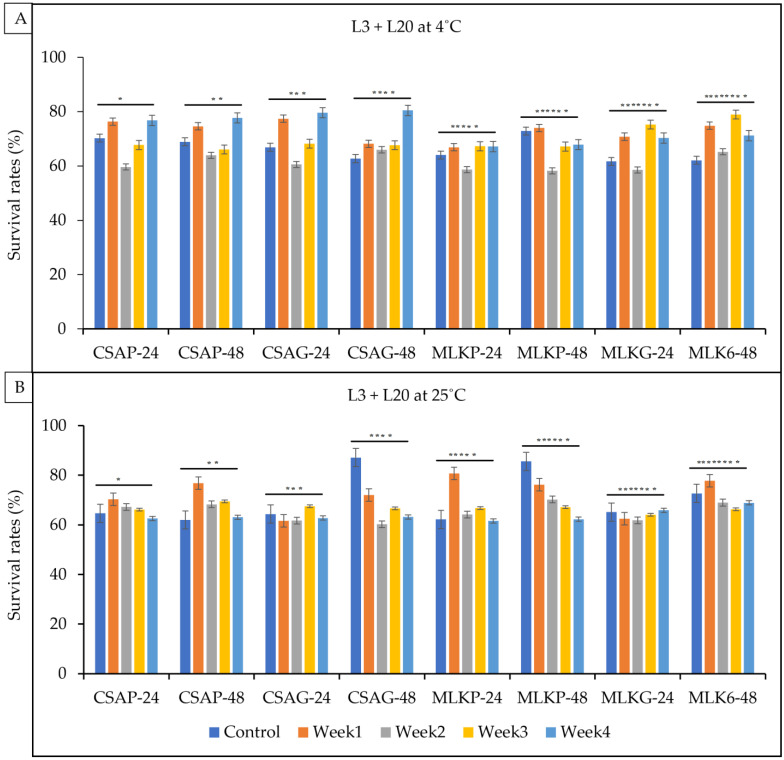
Viability of *Lactococcus lactis* (L3) + *Leuconostoc mesenteroides* (L20) following freeze-drying. (**A**) Survival rates at ambient temperature (25 °C), and (**B**) survival rates at refrigerated temperature (4 °C) for 24 h and 48 h. CSAP-24-Cassava in plastic at 24 h, CSAP-48-Cassava in plastic at 48 h, CSAG-24-Cassava in glass at 24 h, CSAG-48-Cassava in glass at 48 h, MLKp-24-Milk in plastic at 24 h, MLKP-48-Milk in plastic at 48 h, MLKG-24-Milk in glass at 24 h, MLKG-48-Milk in glass at 48 h. (Values represent means of three replicate experiments, ±: standard deviation). For (**A**) * *p*-value = 0.35322, ** *p*-value = 0.24439, *** *p*-value = 0.06101, **** *p*-value = 0.00014, ***** *p*-value = 0.27411, ****** *p*-value = 0.00294, ******* *p*-value = 0.00228, ******** *p*-value = 0.00001. For (**B**) * *p*-value = 0.02542, ** *p*-value = 0.00010, *** *p*-value = 0.09876, **** *p*-value = 0.000000, ***** *p*-value = 0.00714, ****** *p*-value = 0.000001, ******* *p*-value = 0.00857, ******** *p*-value = 0.05367.

**Figure 7 foods-12-01207-f007:**
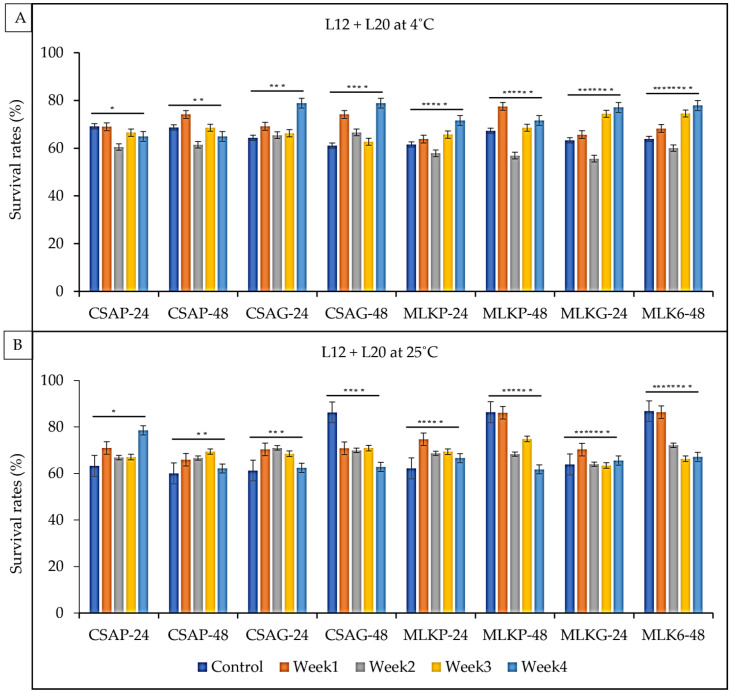
Viability of *Lactobacillus delbrueckii* (L12) + *Leuconostoc mesenteroides* (L20) following freeze-drying. (**A**) Survival rates at ambient temperature (25 °C), and (**B**) survival rates at refrigerated temperature (4 °C) for 24 h and 48 h. CSAP-24-Cassava in plastic at 24 h, CSAP-48-Cassava in plastic at 48 h, CSAG-24-Cassava in glass at 24 h, CSAG-48-Cassava in glass at 48 h, MLKp-24-Milk in plastic at 24 h, MLKP-48-Milk in plastic at 48 h, MLKG-24-Milk in glass at 24 h, MLKG-48-Milk in glass at 48 h (Values represent means of three replicate experiments, ±: standard deviation). For (**A**) * *p*-value = 0.00053, ** *p*-value = 0.00000, *** *p*-value = 0.00002, **** *p*-value = 0.000000, ***** *p*-value = 0.000080, ****** *p*-value = 0.000383, ******* *p*-value = 0.02219, ******** *p*-value = 0.000086. For (**B**) * *p*-value = 0.000940, ** *p*-value = 0.125886, *** *p*-value = 0.001581, **** *p*-value = 0.000159, ***** *p*-value = 0.03362, ****** *p*-value = 0.42607, ******* *p*-value = 0.08091, ******** *p*-value = 0.007825.

**Figure 8 foods-12-01207-f008:**
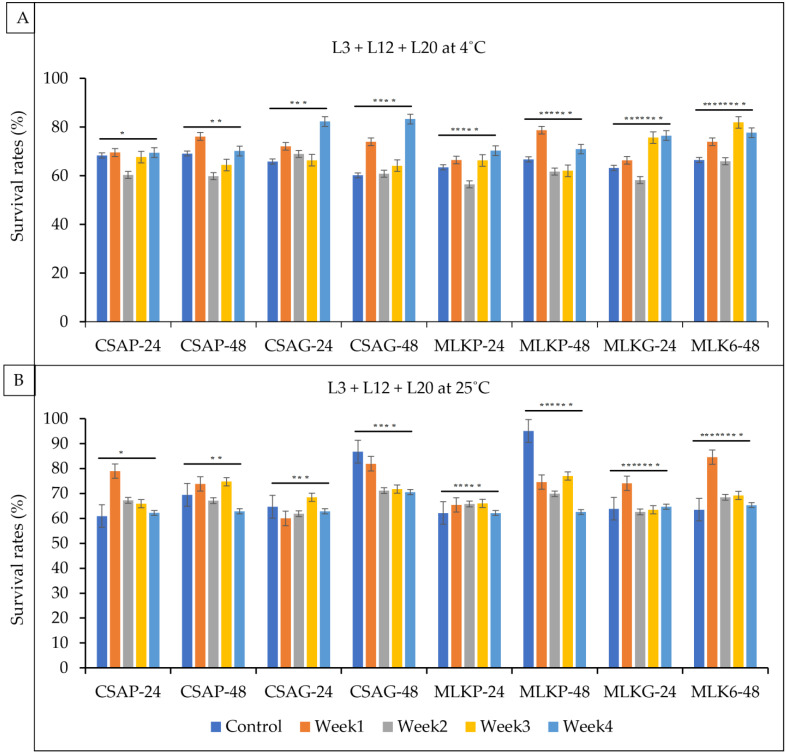
Viability of *Lactococcus lactis* (L3) + *Lactobacillus delbrueckii* (L12) + *Leuconostoc mesenteroides* (L20) following freeze-drying. (**A**) Survival rates at ambient temperature (25 °C), and (**B**) survival rates at refrigerated temperature (4 °C) for 24 h and 48 h. CSAP-24-Cassava in plastic at 24 h, CSAP-48-Cassava in plastic at 48 h, CSAG-24-Cassava in glass at 24 h, CSAG-48-Cassava in glass at 48 h, MLKp-24-Milk in plastic at 24 h, MLKP-48-Milk in plastic at 48 h, MLKG-24-Milk in glass at 24 h, MLKG-48-Milk in glass at 48 h. (Values represent means of three replicate experiments, ±: standard deviation). For (**A**) * *p*-value = 0.083647, ** *p*-value = 0.149906, *** *p*-value = 0.001165, **** *p*-value = 0.000686, ***** *p*-value = 0.290059, ****** *p*-value = 0.318028, ******* *p*-value = 0.020805, ******** *p*-value = 0.000277. For (**B**) * *p*-value = 0.000436, ** *p*-value = 0.459237, *** *p*-value = 0.439752, **** *p*-value = 0.000001, ***** *p*-value = 0.000397, ****** *p*-value = 0.000000, ******* *p*-value = 0.065586, ******** *p*-value = 0.000688.

**Figure 9 foods-12-01207-f009:**
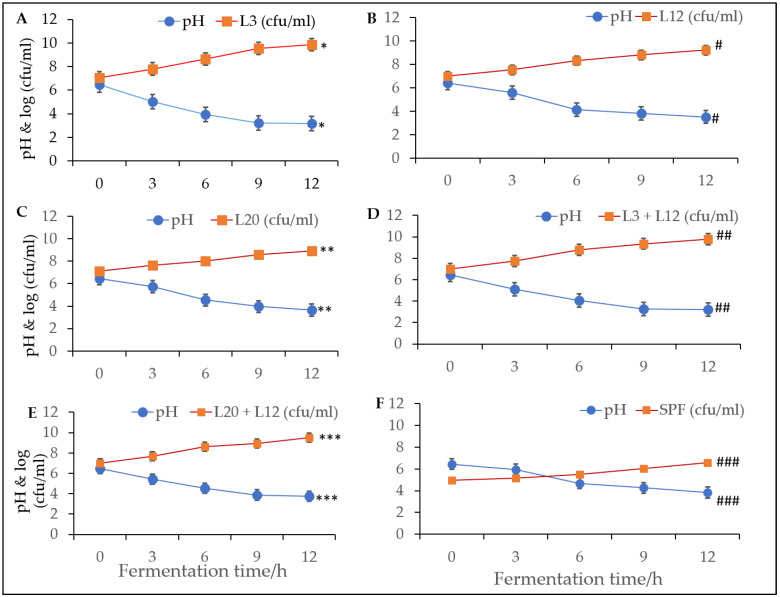
LAB count (log cfu/mL) and pH of yogurt produced from freeze-dried starter cultures. (**A**) Growth trajectory of *Lactococcus lactis* (L3) and pH under increasing fermentation time. (**B**) Growth trajectory of *Lactobacillus delbrueckii* (L12) and pH under increasing fermentation time. (**C**) Growth trajectory of *Leuconostoc mesenterides* (L20) and pH under increasing fermentation time. (**D**) Growth trajectory of *Lactococcus lactis* (L3) combined with *Leuconostoc mesenterides* (L20) and pH under increasing fermentation time. (**E**) Growth trajectory of *Lactobacillus delbrueckii* (L12) combined with *Leuconostoc mesenterides* (L20) and pH under increasing fermentation time. (**F**) Growth trajectory of Spontaneous fermentation (SPF) bacteria and pH under increasing fermentation time. Colony forming units (cfu). For (**A**), * *p*-value = 0.0000983. For (**B**), # *p*-value = 0.000098. For (**C**), ** *p*-value = 0.000208. For (**D**), ## *p*-value = 0.000099. For (**E**), *** *p*-value = 0.0001052. For (**F**), ### *p*-value = 0.042738.

**Table 1 foods-12-01207-t001:** Single and combined freeze-dried starter cultures on fermentation.

Fermentation	Starter Culture	Codes of Starter Culture
Single starter culture	*Lactococcus lactis*	L3
	*L. delbrueckii subsp. bulgaricus*	L12
	*Leuconostoc mesenteroides*	L20
Combined starter cultures	*Lactococcus lactis* + *L. delbrueckii*	L3 + L12
	*Leuconostoc mesenteroides* + *L. delbrueckii*	L20 + L12
Spontaneous fermentation (control)	No starter added	SPF

**Table 2 foods-12-01207-t002:** Consumer sensory evaluation of traditional yogurts produced with freeze-dried starter cultures.

Starter Culture	Sensory Attribute				
	Color	Odor	Taste	Texture	Overall Acceptability
L3	8.04 ± 1.35 ^a^	7.16 ± 1.21 ^a^	7.23 ± 1.30 ^a^	7.72 ± 1.52 ^a^	8.36 ± 1.18 ^a^
L12	7.95 ± 1.40 ^a^	7.01 ± 1.19 ^a^	6.65 ± 1.25 ^b^	7.60 ± 1.45 ^a^	7.22 ± 1.15 ^b^
L20	8.00 ± 1.20 ^a^	8.37 ± 1.25 ^b^	7.14 ± 1.30 ^ac^	7.84 ± 1.33 ^ab^	7.39 ± 1.50 ^b^
L3 + L12	8.06 ± 0.95 ^a^	8.83 ± 1.10 ^b^	7.37 ± 1.05 ^a^	8.08 ± 0.96 ^b^	8.45 ± 0.99 ^a^
L20 + L12	7.90 ± 1.15 ^a^	7.20 ± 1.21 ^a^	6.91 ± 1.27 ^c^	7.75 ± 1.65 ^a^	7.30 ± 1.05 ^b^
SPF	7.96 ± 1.20 ^a^	6.03 ± 0.83 ^c^	6.20 ± 1.24 ^d^	7.02 ± 0.73 ^c^	6.25 ± 1.00 ^c^

Values represent means of three independent experiments, ±: standard deviation. Values in the same column with different superscript letters are significantly different from each other (*p* < 0.05). L3: *Lactococcus lactis*; L12: *Lactobacillus delbrueckii*; L20: *Leuconostoc mesenteroides*; L3 + L20: combined starter of *Lact. lactis* and *Leuc. mesenteroides*; L20 + L12: combined starter of *Leuc. mesenteroides* and *Lact. lactis*; SPF: spontaneous fermentation (without starter culture).

## Data Availability

All data are contained within the article.

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
