# Peer review of "A Comparative Study of Skimmed Milk and Cassava Flour on the Viability of Freeze-Dried Lactic Acid Bacteria as Starter Cultures for Yogurt Fermentation"

_foods, 2023, doi:10.3390/foods12061207_

Round 1

Reviewer 1 Report

This paper was evaluate the survival rates of lactic acid bacteria cultures isolated from Ghanaian fermented milk following freeze-drying, and their performance for milk fermentation. The results reveal the potent of indigenous single or combine lactic acid bacteria starter cultures for milk fermentation in Ghana.

General concept comments
o The greatest weakness is raised by the use of OD for viability of LAB following freeze-drying as shown in figures 2-8. Measurement of optical density includes both viable and non-viable cells. The authors did not provide any results of viable cell count (CFU/g) and survival rate of LAB following freeze drying and storage time for all treatments

Comment 1:
o Figures 2-8 show the viability of LAB following freeze-drying in OD unit which are not in-line with the text in the result that mention cell viability as % of survival rates. It is suggested to change to viable cell in CFU/g instead of OD.

Comment 2:
o Line 377-378 (Figure 9): The figure is not professionally presented. The names and units are not written in X-axis. Is it fermentation time in hour?? There is not any explanation of L3, L12 and L20, below the figure 9.

Comment 3:
o Line 373-374: Lactococcus lactis recorded the lowest pH rate as well as the highest CFU/mL. Is it significantly different to others?. Add the values.

Comment 4:
o Line 539-541: All freeze-dried lactic acid bacteria starter cultures whether single or in combinations grew rapidly during yogurt making, reducing the pH of milk to below 4 units within 9 hours of fermentation. There are not any data in the result section that support this statement.

Author Response

Reviewer 1

General concept comments
o The greatest weakness is raised by the use of OD for viability of LAB following freeze-drying as shown in figures 2-8. Measurement of optical density includes both viable and non-viable cells. The authors did not provide any results of viable cell count (CFU/g) and survival rate of LAB following freeze drying and storage time for all treatments

Response: Although we measure using OD for viabilities, the strains that were selected for the fermentation process viability were measured by employing the everyday cfu/ml technique, and re-confirmed by viability using OD. Their ability to further cause fermentation was an additional proof that the cells were indeed viable.

Comment 1:
o Figures 2-8 show the viability of LAB following freeze-drying in OD unit which are not in-line with the text in the result that mention cell viability as % of survival rates. It is suggested to change to viable cell in CFU/g instead of OD.

Response: All figures are converted to survival rates.

Comment 2:
o Line 377-378 (Figure 9): The figure is not professionally presented. The names and units are not written in X-axis. Is it fermentation time in hour?? There is not any explanation of L3, L12 and L20, below the figure 9.

Response: Graphs were revised professionally, and figures well described.

Comment 3:
o Line 373-374: Lactococcus lactis recorded the lowest pH rate as well as the highest CFU/mL. Is it significantly different to others? Add the values.

Response: Significant differences indicated.

Comment 4:
o Line 539-541: All freeze-dried lactic acid bacteria starter cultures whether single or in combinations grew rapidly during yogurt making, reducing the pH of milk to below 4 units within 9 hours of fermentation. There are not any data in the result section that support this statement.

Response: Statement reconstructed.

Reviewer 2 Report

Dear Editor, thank you for inviting me to review the manuscript:

A comparative study of skimmed milk and cassava flour on the viability of freeze-dried lactic acid bacteria as starter cultures for yogurt fermentation

1.      First of all, Authors must review and revise their manuscript to comply with the requirements of this Journal (spaces, table captions, figures, references to figures in the text, incorrect entries in References).

2.      The abstract is too long. Authors should focus on the most important results and conclusions.

3.      I have serious doubts whether literature 9, 10, 11, 12 is correctly cited. Quoting comes up quite often. Please verify.

4.      Line 179: “Leuconostoc” not „euconostoc

5.      Chapter 2.9. I have a few doubts about this chapter: Were the panelists pre-trained? How were the panelists selected? What was their age? Analysis conditions? Did the authors receive approval from the Ethics Committee for such research?

6.      Line 432: Only in Africa? Needs to be improved.

7.      The Results and Discussion chapters are hard to read. Coherence is missing. In the Discussion chapter, the authors should also refer to the obtained results and compare them with the reports of other researchers.

Author Response

Reviewer 2

Dear Editor, thank you for inviting me to review the manuscript:

A comparative study of skimmed milk and cassava flour on the viability of freeze-dried lactic acid bacteria as starter cultures for yogurt fermentation

  1. First of all, Authors must review and revise their manuscript to comply with the requirements of this Journal (spaces, table captions, figures, references to figures in the text, incorrect entries in References).
  2. The abstract is too long. Authors should focus on the most important results and conclusions.

Response: Abstract revised accordingly.

  1. I have serious doubts whether literature 9, 10, 11, 12 is correctly cited. Quoting comes up quite often. Please verify.

Response: Literature 9, 10, 11, 12 were verified to be correctly cited. It is repeated in the introduction about 3 times. Generally, literature has been extensively revised.

  1. Line 179: “Leuconostoc” not „euconostoc

Response: Corrected.

  1. Chapter 2.9. I have a few doubts about this chapter: Were the panelists pre-trained? How were the panelists selected? What was their age? Analysis conditions? Did the authors receive approval from the Ethics Committee for such research?

Response: The panelists were untrained but were familiar with the products. Ethical statement is put in the text now.

  1. Line 432: Only in Africa? Needs to be improved.

Response: Formatted accordingly.

  1. The Results and Discussion chapters are hard to read. Coherence is missing. In the Discussion chapter, the authors should also refer to the obtained results and compare them with the reports of other researchers.

Response: Data modified extensively with little modifications to the discussion chapter.

Reviewer 3 Report

Dear authors and editors:

The authors have done good work entitled “A Comparative Study of Skimmed Milk and Cassava Flour on the Viability of Freeze-Dried Lactic Acid Bacteria as Starter Cultures for Yogurt Fermentation”. I have some comments that should be done to improve your manuscript before considering it for publication in a good journal such as “Foods” journal.

·       First of all, please check the manuscript and follow the Foods journal format. Also, some grammar mistakes should be revised. Please check and revise the whole manuscript.

·       In the affiliation, please check the authors affiliation number.

·       The abstract is too long, hope the author follows this structure and writes the abstract in brief, including background and problem, the rationale for the study; research objectives; some methodology; important data including statistical analysis; conclusions, novelty, and the importance of the findings.

·       I would the authors to add more keywords with more details.

·       The introduction should have three main points 1) Skimmed milk and Cassava flour, 2) the Freeze-dried method, and 3) Starter cultures.

·       In the introduction, the authors should add three paragraphs more, (1) what are the benefits of the lactic acid bacteria in the fermentation in a single culture or combine culture as well mention why they selected those starters (add some information about the starter). (2) the authors should add some information about the application of the freeze-drying methods compared to other methods and why they preferred this method (freeze-drying). (3) the materials they applied in the freeze-drying method (why they used these materials).

·       In the whole manuscript, the authors should write the scientific names of the starters or the microorganisms in italic form. Please check and revise the whole manuscript.

·       Rewrite the aim of your work clearly with more details and how it is good for a food science major. Could the authors also explain the reason for selecting the Lactobacillus delbrueckii, Lactococcus lactis and Leuconostoc mesenteroides for their study?

·       In the study design, what about the ratio between the starter in single and in a combined and the wall material (Skimmed milk and Cassava flour)?

·       In figure 1, the authors should add the Temp. and time for the freeze-drying. As well as add the ratio between the starter and wall material.

·       In the section on lactic acid bacteria strains, did the authors isolate and identify them in previous studies? Please mention the references

·       Also, did the authors study the safety of the isolated strain of bacteria?

·       How many times did the authors wash the cells in the phosphate buffer solution?

·       Is there any standard for ambient temperature and evaluation time for the whole Consumer sensory evaluation of fermented milk? Please add this information.

·       For all the figures, the quality should be improved, for each sample, they should use a different color, and they should add significant letters among the samples in each treatment.

·       In the all figures captions add the abbreviation meaning and the meaning of A, B, C, …

·       For the abbreviation of the hour, sometimes they used hrs and h, please follow one in the whole manuscript.

·       Line 275, after what?

·       Appendix A?

·       Appendix B?

·       References, please revise the format

Author Response

Reviewer 3

Dear authors and editors:

The authors have done good work entitled “A Comparative Study of Skimmed Milk and Cassava Flour on the Viability of Freeze-Dried Lactic Acid Bacteria as Starter Cultures for Yogurt Fermentation”. I have some comments that should be done to improve your manuscript before considering it for publication in a good journal such as “Foods” journal.

  • First of all, please check the manuscript and follow the Foods journal format. Also, some grammar mistakes should be revised. Please check and revise the whole manuscript.

Response: English language has been duly edited, Foods Journal format has been used to revise the manuscript. Entire manuscript have been revised.

  • In the affiliation, please check the authors affiliation number.

Response: Author’s affiliation numbers has been checked.

  • The abstract is too long, hope the author follows this structure and writes the abstract in brief, including background and problem, the rationale for the study; research objectives; some methodology; important data including statistical analysis; conclusions, novelty, and the importance of the findings.

Response: Abstract have been edited accordingly.

  • I would the authors to add more keywords with more details.

Response: More keywords added

  • The introduction should have three main points 1) Skimmed milk and Cassava flour, 2) the Freeze-dried method, and 3) Starter cultures.

Introduction revised.

Response: This information is contained in our introduction.

  • In the introduction, the authors should add three paragraphs more, (1) what are the benefits of the lactic acid bacteria in the fermentation in a single culture or combine culture as well mention why they selected those starters (add some information about the starter).

Response: We have included these in the revised manuscript.

(2) the authors should add some information about the application of the freeze-drying methods compared to other methods and why they preferred this method (freeze-drying).

Response: The application has been highlighted in the revised manuscript

(3) the materials they applied in the freeze-drying method (why they used these materials).

  • In the whole manuscript, the authors should write the scientific names of the starters or the microorganisms in italic form. Please check and revise the whole manuscript.

Response: Bacterial strains used have been italicized.

  • Rewrite the aim of your work clearly with more details and how it is good for a food science major. Could the authors also explain the reason for selecting the Lactobacillus delbrueckiiLactococcus lactisand Leuconostoc mesenteroides for their study?

Response: Revised accordingly

  • In the study design, what about the ratio between the starter in single and in a combined and the wall material (Skimmed milk and Cassava flour)?

Response: The ratios have been captured as specific quantity used in the study for verification’s sake. Ratios might be determined using such specific figures.

  • In figure 1, the authors should add the Temp. and time for the freeze-drying. As well as add the ratio between the starter and wall material.

Response: Revised accordingly, however, some parameters have been clearly stated in the methodology write up.

  • In the section on lactic acid bacteria strains, did the authors isolate and identify them in previous studies? Please mention the references

Response: Revised accordingly, reference of isolates provided.

  • Also, did the authors study the safety of the isolated strain of bacteria?

Response: This bacteria has been well studied, and the WHO and CDC categorizes these bacteria as GRAS (Generally Regarded As Safe), and this has been stated in the work.

  • How many times did the authors wash the cells in the phosphate buffer solution?

Response: twice.

  • Is there any standard for ambient temperature and evaluation time for the whole Consumer sensory evaluation of fermented milk? Please add this information.

Response: No please. Ambient temperature varies depending on the location.

  • For all the figures, the quality should be improved, for each sample, they should use a different color, and they should add significant letters among the samples in each treatment.

Response: Figures revised to improve quality.

  • In the all figures captions add the abbreviation meaning and the meaning of A, B, C, …

Response: Revised accordingly.

  • For the abbreviation of the hour, sometimes they used hrs and h, please follow one in the whole manuscript.

Response: Revised to have hour as h. Revision made throughout the manuscript.

  • Line 275, after what?
  • Appendix A?
  • Appendix B?

Response: Appendix A and B removed. Editorial error.

  • References, please revise the format

Response: Reference made using the Foods Journal Format.

Round 2

Reviewer 1 Report

This revised version of the manuscript is accepted

Author Response

Reviewer 1

Review Report Form

Open Review

English language and style

( ) English very difficult to understand/incomprehensible
( ) Extensive editing of English language and style required
( ) Moderate English changes required
(x) English language and style are fine/minor spell check required
( ) I don't feel qualified to judge about the English language and style

Yes

Can be improved

Must be improved

Not applicable

Does the introduction provide sufficient background and include all relevant references?

(x)

( )

( )

( )

Are all the cited references relevant to the research?

(x)

( )

( )

( )

Is the research design appropriate?

(x)

( )

( )

( )

Are the methods adequately described?

(x)

( )

( )

( )

Are the results clearly presented?

(x)

( )

( )

( )

Are the conclusions supported by the results?

(x)

( )

( )

( )

Comments and Suggestions for Authors

This revised version of the manuscript is accepted

Response: We sincerely thank reviewer for making time out of his/her busy schedules to go through our manuscript. The comments have greatly enhanced the manuscript, and we appreciate it. We are also very excited that he/she have accepted this manuscript.  

Submission Date

15 January 2023

Date of this review

06 Feb 2023 01:53:17

Reviewer 2 Report

Now, the manuscript looks better.

Author Response

Reviewer 2

Review Report Form

Open Review

English language and style

( ) English very difficult to understand/incomprehensible
( ) Extensive editing of English language and style required
( ) Moderate English changes required
(x) English language and style are fine/minor spell check required
( ) I don't feel qualified to judge about the English language and style

Yes

Can be improved

Must be improved

Not applicable

Does the introduction provide sufficient background and include all relevant references?

(x)

( )

( )

( )

Are all the cited references relevant to the research?

(x)

( )

( )

( )

Is the research design appropriate?

(x)

( )

( )

( )

Are the methods adequately described?

(x)

( )

( )

( )

Are the results clearly presented?

(x)

( )

( )

( )

Are the conclusions supported by the results?

(x)

( )

( )

( )

Comments and Suggestions for Authors

Now, the manuscript looks better.

Response: We sincerely thank reviewer for making time out of his/her busy schedules to go through our manuscript. The comments have greatly enhanced the manuscript, and we appreciate it. We are also very excited that he/she have accepted this manuscript.  

Submission Date

15 January 2023

Date of this review

03 Feb 2023 12:54:55

Reviewer 3 Report

Dear Authors, please revise the manuscript format (title and subtitle, and references (the journals should be in abbreviation and italics) as well as highlight the changes of my points. Some of them, you did and some you skip. Please revise it carefully. thanks

Author Response

Reviewer 3

Dear authors and editors:

The authors have done good work entitled “A Comparative Study of Skimmed Milk and Cassava Flour on the Viability of Freeze-Dried Lactic Acid Bacteria as Starter Cultures for Yogurt Fermentation”. I have some comments that should be done to improve your manuscript before considering it for publication in a good journal such as “Foods” journal.

  • First of all, please check the manuscript and follow the Foods journal format. Also, some grammar mistakes should be revised. Please check and revise the whole manuscript.

Response: English language has been duly edited; Foods Journal format has been used to revise the manuscript. Entire manuscript have been revised.

  • In the affiliation, please check the authors affiliation number.

Response: Author’s affiliation numbers has been checked.

  • The abstract is too long, hope the author follows this structure and writes the abstract in brief, including background and problem, the rationale for the study; research objectives; some methodology; important data including statistical analysis; conclusions, novelty, and the importance of the findings.

Response: Abstract have been edited accordingly.

Response: Abstract have been revised into only 248 words: Thank you and this has greatly improved our manuscript.

  • I would the authors to add more keywords with more details.

Response: More keywords added

  • The introduction should have three main points 1) Skimmed milk and Cassava flour, 2) the Freeze-dried method, and 3) Starter cultures.

Introduction revised.

Response: This information is contained in our introduction.

  • In the introduction, the authors should add three paragraphs more, (1) what are the benefits of the lactic acid bacteria in the fermentation in a single culture or combine culture as well mention why they selected those starters (add some information about the starter).

Response: We have included these in the revised manuscript.

(2) the authors should add some information about the application of the freeze-drying methods compared to other methods and why they preferred this method (freeze-drying).

Response: The application has been highlighted in the revised manuscript

(3) the materials they applied in the freeze-drying method (why they used these materials).

  • In the whole manuscript, the authors should write the scientific names of the starters or the microorganisms in italic form. Please check and revise the whole manuscript.

Response: Bacterial strains used have been italicized.

  • Rewrite the aim of your work clearly with more details and how it is good for a food science major. Could the authors also explain the reason for selecting the Lactobacillus delbrueckiiLactococcus lactisand Leuconostoc mesenteroides for their study?

Response: Revised accordingly

  • In the study design, what about the ratio between the starter in single and in a combined and the wall material (Skimmed milk and Cassava flour)?

Response: The ratios have been captured as specific quantity used in the study for verification’s sake. Ratios might be determined using such specific figures.

  • In figure 1, the authors should add the Temp. and time for the freeze-drying. As well as add the ratio between the starter and wall material.

Response: Revised accordingly, however, some parameters have been clearly stated in the methodology write up.

  • In the section on lactic acid bacteria strains, did the authors isolate and identify them in previous studies? Please mention the references

Response: Revised accordingly, reference of isolates provided.

  • Also, did the authors study the safety of the isolated strain of bacteria?

Response: These bacterial strains have been well studied, and the WHO and CDC categorizes these bacteria as GRAS (Generally Regarded As Safe), and this has been stated in the work.

  • How many times did the authors wash the cells in the phosphate buffer solution?

Response: twice.

  • Is there any standard for ambient temperature and evaluation time for the whole Consumer sensory evaluation of fermented milk? Please add this information.

Response: No please. Ambient temperature varies depending on the location.

  • For all the figures, the quality should be improved, for each sample, they should use a different color, and they should add significant letters among the samples in each treatment.

Response: Figures revised to improve quality.

  • In all figures captions add the abbreviation meaning and the meaning of A, B, C, …

Response: All figure legends have been revised, and A, B, C, D etc. labelled appropriately.

  • For the abbreviation of the hour, sometimes they used hrs and h, please follow one in the whole manuscript.

Response: Revised to have hour as h. Revision made throughout the manuscript.

Response: Document have been revised to remove hrs and replace h for hours.

  • Line 275, after what?
  • Appendix A?
  • Appendix B?

Response: Appendix A and B removed. Editorial error.

  • References, please revise the format

Response: Reference made using the Foods Journal Format.

Response: References have been duly revised to conform to Foods format

Author's Notes File

Report Notes

Review Report Form

Open Review

English language and style

( ) English very difficult to understand/incomprehensible
( ) Extensive editing of English language and style required
( ) Moderate English changes required
(x) English language and style are fine/minor spell check required
( ) I don't feel qualified to judge about the English language and style

Yes

Can be improved

Must be improved

Not applicable

Does the introduction provide sufficient background and include all relevant references?

( )

(x)

( )

( )

Are all the cited references relevant to the research?

( )

(x)

( )

( )

Is the research design appropriate?

( )

( )

( )

( )

Are the methods adequately described?

( )

(x)

( )

( )

Are the results clearly presented?

( )

(x)

( )

( )

Are the conclusions supported by the results?

( )

(x)

( )

( )

Comments and Suggestions for Authors

Dear Authors, please revise the manuscript format (title and subtitle, and references (the journals should be in abbreviation and italics) as well as highlight the changes of my points. Some of them, you did and some you skip. Please revise it carefully. Thanks.

Response:

  • All titles have been numbered and boldened.
  • All subtitles have italicized.
  • References has been changed to conform to Foods Format
  • Also, abstract has been revised into 248 words only.  

Response: We sincerely thank reviewer for making time out of his/her busy schedules to go through our manuscript. The comments have greatly enhanced the manuscript, and we appreciate it.

Submission Date

15 January 2023

Date of this review

02 Feb 2023 13:49:04
